The invasive New Guinea flatworm Platydemus manokwari in France, the first record for Europe: time for action is now

Justine Jean-Lou 1 justine@mnhn.fr
Winsor Leigh 2
Gey Delphine 3
Gros Pierre 4
Thévenot Jessica 5
1 ISYEB, Institut de Systématique Evolution Biodiversité, UMR7205 CNRS, EPHE, MNHN, UPMC, Muséum National d’Histoire Naturelle , France
2 School of Marine and Tropical Biology, James Cook University , Australia
3 Service de Systématique moléculaire, Muséum National d’Histoire Naturelle , France
4 Amateur Naturalist , Cagnes-sur-Mer , France
5 Coordination technique et scientifique de la stratégie nationale relative aux espèces exotiques envahissantes, Service du Patrimoine Naturel, Muséum National d’Histoire Naturelle , France
Grandcolas Philippe
Electronic publication date: 2014 Mar 4
Publication date: 2014
Volume: 2
Electronic Location ID: e297
Received 2014 Jan 20; Accepted 2014 Feb 10
Copyright: © 2014 Justine et al.
Copyright year: 2014
Copyright holder: Justine et al.
License: This is an open access article distributed under the terms of the Creative Commons Attribution License, which permits unrestricted use, distribution, reproduction and adaptation in any medium and for any purpose provided that it is properly attributed. For attribution, the original author(s), title, publication source (PeerJ) and either DOI or URL of the article must be cited.
License URL: https://creativecommons.org/licenses/by/4.0/

Keywords: Invasive species, Land planarian, Platyhelminthes, France, Invader, Biodiversity, Europe, Alien species, Threat to biodiversity

Funding: MNHN ATM “Barcode” MNHN’s ATM “Barcode” provided support for molecular analysis. The funders had no role in study design, data collection and analysis, decision to publish, or preparation of the manuscript.

==============================
Non-indigenous terrestrial flatworms (Platyhelminthes) have been recorded in thirteen European countries. They include Bipalium kewense and Dolichoplana striata that are largely restricted to hothouses and may be regarded as non-invasive species. In addition there are species from the southern hemisphere such as the invasive New Zealand flatworm Arthurdendyus triangulatus in the United Kingdom, Eire and the Faroe Islands, the Australian flatworm Australoplana sanguinea alba in Eire and the United Kingdom, and the Australian Blue Garden flatworm Caenoplana coerulea in France, Menorca and the United Kingdom. The United Kingdom has some twelve or more non-indigenous species most of which are Australian and New Zealand species. These species may move to an invasive stage when optimum environmental and other conditions occur, and the flatworms then have the potential to cause economic or environmental harm. In this paper, we report the identification (from morphology and molecular analysis of COI sequences) of non-indigenous terrestrial flatworms found in a hothouse in Caen (France) as the New Guinea flatworm Platydemus manokwari de Beauchamp, 1963 (Platyhelminthes, Continenticola, Geoplanidae, Rhynchodeminae). Platydemus manokwari is among the “100 World’s Worst Invader Alien Species”. Lists of World geographic records, prey in the field and prey in laboratories of P. manokwari are provided. This species is considered a threat to native snails wherever it is introduced. The recent discovery of P. manokwari in France represents a significant extension of distribution of this Invasive Alien Species from the Indo-Pacific region to Europe. If it escaped the hothouse, the flatworm might survive winters and become established in temperate countries. The existence of this species in France requires an early warning of this incursion to State and European Union authorities, followed by the eradication of the flatworm in its locality, tightening of internal quarantine measures to prevent further spread of the flatworm to and from this site, identifying if possible the likely primary source of the flatworm, and tracing other possible incursions that may have resulted from accidental dispersal of plants and soil from the site.

Introduction

An undesirable consequence of globalization, a relatively modern phenomenon, has been an increase in the number of biological invasions that challenge the conservation of biodiversity and natural resources (Secretariat of NOBANIS, 2012; Simberloff, 2014). Invasive Alien Species (IAS) have been defined as “plants, animals, pathogens and other organisms that are non-native to an ecosystem, and which may cause economic or environmental harm or adversely affect human health. In particular, they impact adversely upon biodiversity, including decline or elimination of native species–through competition, predation, or transmission of pathogens–and the disruption of local ecosystems and ecosystem functions” (Convention on Biological Diversity, 2009).

Historic biological invasions include the passive dispersal of terrestrial flatworms, also known as land planarians. The main driver for this was probably horticulturalists of the 19th Century using the then recently invented Wardian cases to safely transport back to the hothouses and gardens of Europe rare plants, together with soil containing cryptic exotic animal species (Winsor, Johns & Barker, 2004). As a consequence, over 30 species of land planarians have established themselves as non-indigenous species in various countries outside their native range (Winsor, Johns & Barker, 2004).

In human-modified habitat flatworms and their cocoons continue to be associated with rooted and potted plants, rhizomes, and certain types of fresh vegetable produce (Alford, Lole & Emmett, 1996). Subsequent secondary dispersal of these invasive flatworm species occurs through the exchange and purchase of plants from nurseries, botanical gardens, garden centres and gardeners (Alford, Lole & Emmett, 1996) especially infested nurseries and garden centres (Boag et al., 1994; Moore, Dynes & Murchie, 1998; Boag & Yeates, 2001), active inadvertent dispersal through social traditions of exchanging plants and recycling topsoil (Christensen & Mather, 1998), or through the deliberate introduction of flatworms for the purposes of biological control of a pest species such as the giant African snail Achatina fulica Bowdich, 1822 in the Pacific Region (Muniappan, 1987; Waterhouse & Norris, 1987).

Land planarians are carnivores, and feed upon a variety of soil organisms such as earthworms, isopods, insects and snails, and some IAS flatworms may pose a threat to local biodiversity (Alford, Lole & Emmett, 1996; Cowie, 2010; Santoro & Jones, 2001; Ducey, McCormick & Davidson, 2007; Sugiura, 2010) and negatively impact on agriculture, for example through a decline in earthworms species (Murchie & Gordon, 2013) resulting in reduced soil fertility (Murchie, 2010) and possibly drainage (Jones et al., 2001).

Non-indigenous terrestrial flatworms have been recorded in thirteen European countries (Filella-Subirà, 1983; Jones, 1998; Kawakatsu et al., 2002; Ogren, Kawakatsu & Froehlich, 1997). These flatworms can be divided into two broad groups: the “old” and the “new” introduced species.

The “old” group includes Bipalium kewense Moseley, 1878 and Dolichoplana striata Moseley, 1877 that were undoubtedly inadvertently introduced to Europe in the 19th Century by horticulturalists. One or both these species are present in ten European countries, and are the only non-indigenous flatworms presently recorded in Austria, Belgium, Czech Republic, Finland, Germany, Norway, Poland, and Portugal. These two species appear to be largely restricted to hothouses in Europe and do not meet the foregoing criteria for Invasive Alien Species; the species are widespread but exist in localized populations, and may be better regarded as non-invasive species. However such species may move to an invasive stage when optimum environmental and other conditions for the species occur, and the flatworms then have the potential to impact on soil fauna, especially earthworms as has occurred in areas of North America with Dolichoplana striata (Hyman, 1954) and Bipalium species (Ducey et al., 2005; Ducey, McCormick & Davidson, 2007).

The “new” group of non-indigenous flatworms present in Europe includes mainly species from the southern hemisphere such as the IAS New Zealand flatworm Arthurdendyus triangulatus (United Kingdom, Eire, Faroe Islands), the Australian flatworm Australoplana sanguinea alba (Moseley, 1877) (Eire, United Kingdom), and the Australian Blue Garden flatworm Caenoplana coerulea Moseley, 1877 (United Kingdom, France, and recently Menorca (Breugelmans et al., 2012) and Spain (Mateos et al., 2013)). The United Kingdom has some twelve or more non-indigenous species most of which are Australian and New Zealand species (Jones, 2005).

We recently identified non-indigenous terrestrial flatworms found in a hothouse in Caen (France) as the New Guinea flatworm Platydemus manokwari de Beauchamp, 1963. The identity of these flatworms was subsequently confirmed by molecular analysis of COI sequences. Platydemus manokwari is among the “100 World’s Worst Invader Alien Species” (Lowe et al., 2000). In this paper, we present evidence for the identification of the species in France, the first record in Europe, and provide a brief review of the records of the species in the world, lists of its known prey, and possible control options.

Material and Methods

Material

Specimens were found in a hothouse in the Jardin des Plantes in Caen (France); according to witnesses, it is likely that similar specimens were present in the hothouse for months. Specimens were collected by hand and sent alive to Paris by postal service. Eight specimens were processed. Five were kept alive and used for prey experiments and detailed photographs; they died after several days and were discarded. Three were killed in hot water and then stored in ethanol (specimens JL81A and JL81B) or formalin (JL81C). A small piece of the body was taken from the two ethanol-fixed individuals for molecular analysis. Photographs were forwarded to one of us (LW) for identification. Histological anatomical investigations were not undertaken at this time. Specimens are deposited in the collections of the Muséum National d’Histoire Naturelle, Paris, under registration number MNHN JL81A–C.

Limited prey experiments were undertaken with the few living available specimens; very simply, flatworms were put in a small plastic container with living snails.

Molecular sequences

Genomic DNA was extracted from a small piece of the worm, using the QIAamp DNA Mini Kit (Qiagen). A fragment of 424 bp of COI gene was amplified with the primers COI-ASmit1 (forward 5′-TTTTTTGGGCATCCTGAGGTTTAT-3′) and COI-ASmit2 (reverse 5′-TAAAGAAAGAACATAATGAAAATG-3′) (Littlewood, Rohde & Clough, 1997). The PCR reaction was performed in 20 µl, containing 1 ng of DNA, 1× CoralLoad PCR buffer, 3 Mm MgCl2, 66 µM of each dNTP, 0.15 µM of each primer, and 0.5 units of Taq DNA polymerase (Qiagen). The amplification protocol was: 4′ at 94°C, followed by 40 cycles of 94°C for 30″, 48°C for 40″, 72°C for 50″, with a final extension at 72°C for 7′. PCR products were purified and sequenced in both directions on 3730xl DNA Analyzer 96-capillary sequencer (Applied Biosystems). Sequences were edited using CodonCode Aligner software (CodonCode Corporation, Dedham, MA, USA), compared to the GenBank database content using BLAST and deposited in GenBank under accession number KF887958. Sequences were compared using MEGA5 (Tamura et al., 2011).

Results

Morphology

The flatworm was broadest in the middle, tapering evenly anteriorly but more abruptly posteriorly (Fig. 1). Two large prominent eyes were situated back from the tip of the elongate snout-like head (Fig. 2). In cross section the flatworm was convex dorsally and flat ventrally. The figured live mature specimen was 50 mm long and 5 mm wide. The mouth was located just behind the midpoint of the ventrum, with gonopore about half way between the mouth and posterior end. The dorsum was a dark olive brown colour, which under a lens showed a fine pale brownish graininess. A pale cream median dorsal longitudinal stripe, some 0.3 mm wide, began just behind the eyes and continued to the posterior tip (Figs. 1 and 3). The olive brown colour graded to grey at the anterior tip. A thin submarginal cream stripe with fine lower greyish margin ran laterally from the anterior end along the length of the body (Fig. 3). The ventral surface was a pale finely mottled light brown (Figs. 4 and 5), slightly paler mid ventrally. These features are consistent with those of Platydemus manokwari de Beauchamp, 1963 (Platyhelminthes, Continenticola, Geoplanidae, Rhynchodeminae) (de Beauchamp, 1962; Kawakatsu, Ogren & Muniappan, 1992; Winsor, 1990).

Figure 1 Platydemus manokwari de Beauchamp, 1963.

Specimen collected in a hothouse, Caen, France. Dorsal view: note median longitudinal line.

Figure 2 Platydemus manokwari de Beauchamp, 1963.

Detail of head, lateral view, showing one of the two slightly protuberant eyes.

Figure 3 Platydemus manokwari de Beauchamp, 1963.

Detail of body, dorsal view, showing pale cream median longitudinal stripe on dark olive brown background.

Figure 4 Platydemus manokwari de Beauchamp, 1963.

Partial ventral view, showing the cream and faint grey marginal stripe, and the creeping sole that is slightly paler along the median line. Scale: millimeters.

Figure 5 Platydemus manokwari de Beauchamp, 1963, experimental predation on indigenous snail.

The flatworm is preying on a snail: it has been disturbed, thus showing the white cylindrical pharynx on the ventral side, protruding and ingesting soft tissues of the snail. The prey is the helicid Eobania vermiculata, a common snail of the Mediterranean region.

Prey experiments

Various living snails were introduced in the same container as a single flatworm. Four out of the 5 specimens died before feeding. A single prey event, on the helicid Eobania vermiculata Müller, 1774 was observed. The cylindrical pharynx, protruding from the ventral surface, was visible when the flatworm was preying on a snail (Fig. 5).

Molecular identification

The two COI sequences we obtained from two individuals were identical. They were compared to the only available COI sequences of a member of the genus Platydemus in GenBank (Platydemus manokwari; Accession number: AF178320.1). The p-distance between our new sequence and the GenBank sequence of Platydemus manokwari was 4%. Although this is not mentioned in the GenBank record, we know that this specimen was collected in Australia by one of us (LW). The population from which the GenBank example of Platydemus manokwari (AF178320.1) was taken had previously been confirmed histologically (by LW from lot LW1065) as P. manokwari and voucher specimens lodged in the Queensland Museum (Registration numbers GL4724, whole specimen in alcohol, and GL4725, 126 slides).

Nomenclatural clarification

There is some variation in the literature about the date of description of P. manokwari, 1962 or 1963. We carefully examined the original publication. The paper was presented at a meeting in December 1962 and is included in the volume dated 1962, but the actual date of publication was April 18, 1963. In accord with Article 21.1 (International Commission on Zoological Nomenclature, 1999), the date of the taxon is 1963. The bibliographical date of the publication remains 1962, but the taxon is Platydemus manokwari de Beauchamp, 1963.

Discussion

Molecular identification

The p-distance between our two sequences and the GenBank sequence of Platydemus manokwari was 4%. This genetic distance roughly corresponds to the genetic distances generally found between closely related species or distant populations within a single species. Alvarez-Presas et al. (2012) studied variation of COI in species belonging to the same family as Platydemus manokwari, i.e., the Geoplanidae. In this study of the European species Microplana terrestris (Muller, 1774), specimens were studied from two localities, East and West of Northern Spain. Variation ranged from 0% to 3% within the western localities, and from 0% to 1.6% in the eastern localities; West and East presented a difference of 2.4% to 4%. The between-species difference (M. terrestris vs M. robusta Vila-Farré and Sluys, 2011) was about 19%. Therefore, we consider that the difference of 4% found between our French specimens of P. manokwari and the Australian specimen in GenBank is compatible with intraspecific variation. The molecular data thus confirm the morphological identification.

Previous records of land planarians in France

Previous records of non-indigenous land planarians in France include a Pelmatoplana sp. from a greenhouse in Saint Max, a suburb of Nancy (identified by de Beauchamp in Remy, 1942); Bipalium kewense and Caenoplana coerulea from an urban garden in Villeneuve-de-la-Raho, department of Pyrénées-Orientales (mentioned as “France” in Winsor, Johns & Barker, 2004), where potted plants purchased from a local plant supermarket were believed to be the source of the flatworms (Gérard Peaucellier, in litt); Bipalium kewense in Orthez and Bayonne, department of Pyrenées-Atlantiques (Vivant, 2005).

Other species, often unidentified, have been recorded recently in France in newspapers (Gasiglia, 2013; Guyon, 2014; Heyligen, 2013), magazines (Groult & Boucourt, 2014) and blogs (Justine, 2013) and mentioned in governmental documents (Placé, 2013), but not in scientific publications.

The occurrence of the invasive flatworm Platydemus manokwari in the Jardin des Plantes, Caen, in the department of Basse-Normandie (Normandy, France), is the first record of the species in Europe.

Previous records of Platydemus manokwari

Platydemus manokwari occurs at Pindaunde station, Mt. Wilhelm at 3625 m altitude (de Beauchamp, 1972; Winsor, 1990), where it was found under stones together with Platydemus longibulbus (de Beauchamp, 1972) and Platydemus pindaudei (de Beauchamp, 1972), and at Kainantu at 1558 m altitude in the eastern highlands of New Guinea (Winsor, 1990). The natural range of this upland species has yet to be determined.

Until now, Platydemus manokwari was confined to the Indo-Pacific region within the bounds of the Ogasawara Islands, Japan in the north; near Mackay in Queensland, Australia to the south; French Polynesia to the east; with the most westerly extent of the flatworm in the Maldives. The Caen record of this species is a significant westerly extension of the occurrence of P. manokwari from the Indo-Pacific region to Europe.

Figure 6 Platydemus manokwari, map of distribution records.

Until now, Platydemus manokwari was confined to the Indo-Pacific region. The present record in France is a significant westerly extension of the occurrence of P. manokwari from the Indo-Pacific region to Europe.

Since it was first discovered in the Agricultural Research Station in Manokwari, Irian Jaya in 1962 where it was credited with the decline of the Giant African Snail Achatina fulica, an invasive pest of coconut plantations and other crops (Mead, 1963; Mead, 1979; Muniappan et al., 1986; Schreurs, 1963; van Driest, 1968; Waterhouse & Norris, 1987), Platydemus manokwari has progressively spread throughout the Indo-Pacific (Table 1, Fig. 6). The flatworm has been accidentally introduced, probably together with plants and soil, to various islands in the Pacific region including Australia, Guam, Palau, Hawaii, Federated States of Micronesia, French Polynesia, and Samoa. The most recent report of P. manokwari in the Pacific region is its occurrence in Rotuma in the Fiji archipelago (Brodie et al., in press). The flatworm was also deliberately introduced as a bio-control agent for the Giant African Snail Achatina fulica to Bugsuk in the Philippines (Muniappan et al., 1986; Waterhouse & Norris, 1987), Yokohama, Japan (Eldredge & Smith, 1995), and the Maldives (Muniappan, 1987). The rate of secondary dispersal of Platydemus manokwari is low and depends upon transport of infected plants and soil, or the flatworms themselves, by humans. The flatworms appear to be incapable of travelling long distances on their own, having taken 12 months to colonize mixed urban garden habitats separated by some 30 m of lawn (Winsor, 1990), and on Fua Mulaku (Maldives) cleared Achatina for a radius of 180 m from the release site over the period of a year (Muniappan, 1987).

Table 1 Platydemus manokwari- Distribution records.

Platydemus manokwari has been recorded from more than 15 different territories, in Asia and Oceania; our record in France is the first for Europe.

Location	Year discovered	Localities, comments	Reference	
Irian Jaya	1962	Agricultural Research Station, Manokwari	de Beauchamp, 1962	
New Guinea	1969	Mt.Wilhelm, Pindaude Station	de Beauchamp, 1972	
	1973	Kainantu, 45 km SE of Goroka	LW identification for C. Vaucher in litt 4.iii.1982	
Australia	1976	Queensland: Lockhart River, Weipa, Atherton Tablelands, Cairns, Mission Beach, Cardwell, Crystal Creek; Bluewater, Townsville.	LW 1985 pers. comm. in Waterhouse & Norris, 1987; Winsor, 1990; Winsor, 1999	
	2002	Northern Territory: Anula	LW identification for C. Glasby in litt 23.x.2002	
	2009	Queensland: Bowen; Airlie Beach	LW collection	
Guam, Northern Mariana Islands	1977–1984	Guam, Saipan, Tinian	Hopper & Smith, 1992; Eldredge & Smith, 1995	
	1988	Rota	Bauman, 1996	
	1992	Aguijan, identified as “Microplaninae sp.”	Eldredge & Smith, 1995; Kawakatsu & Ogren, 1994	
Philippines	1981	Bugsuk, deliberate introduction	Muniappan et al., 1986; Waterhouse & Norris, 1987	
	1985	Manilla, urban	Waterhouse & Norris, 1987	
Japan	1984	Yokohama, deliberate introduction from Saipan	Eldredge & Smith, 1995	
	1990–1991	Okinawa Island and other Ryukyu Islands	Eldredge & Smith, 1995; Kawakatsu et al., 1993	
	1995	Ogasawara (Bonin) Islands	Kawakatsu et al., 1999	
Maldives	1985	Fua Mulaku, deliberate introduction from
Bugsuk and Saipan; Addu Atoll; Male Atoll	Eldredge & Smith, 1995; Muniappan, 1987	
Palau	1991	Koror Island	Eldredge & Smith, 1995	
	1992	Ulong Island	Eldredge & Smith, 1995	
United States - Hawaii	1992	Oahu	Eldredge & Smith, 1995	
Federated States of Micronesia		Pohnpei Ponape Island	Eldredge & Smith, 1995	
French Polynesia	1997	Mangareva Island, road to Mt. Mokoto	LW identification for J. Starmer in litt 21.iii.1998; Purea et al., 1998	
	2009	Moorea	Lovenburg, 2009	
Samoa	1998	Alafua and Upolu	Cowie & Robinson, 2003; Purea et al., 1998	
Tonga			FAO-SAPA, 2002	
Vanuatu			FAO-SAPA, 2002	
Fiji	2012	Rotuma	Brodie, Stevens & Barker, 2012; Brodie et al., in press	
France	2013	Caen, Normandy	This paper	

Reproduction

Under experimental conditions the optimum temperature for rearing P. manokwari in terms of pre-oviposition period and cocoon production is 24°C, with a mean post-oviposition developmental period for the young to hatch from the cocoon of 7.8 ±1.2 days (Kaneda et al., 1992). Cocoons contain an average of 5.2 juveniles (3–9) each. The flatworm begins oviposition within 3 weeks of hatching (Kaneda et al., 1992). The temperature threshold for oviposition lies between 15°C and 18°C, and for cocoon and juvenile stages 10°C and 11.7°C respectively (Kaneda et al., 1992). The flatworm normally reproduces sexually, and does not appear to reproduce by fission (Kaneda, Kitagawa & Ichinohe, 1990).

Biology

Platydemus manokwari prefers wet humid conditions and is unable to survive in completely dry habitats; high humidity and adequate precipitation are essential for the survival of the flatworm (Kaneda, Kitagawa & Ichinohe, 1990; Sugiura, 2009). The flatworm is diurnal if the moisture conditions are right (Kaneda, Kitagawa & Ichinohe, 1990). Temperature appears to influence predation rate by the flatworm in field and laboratory experiments, and also its survival. Sugiura (2009) considers that 10°C is a possible threshold temperature for the establishment of P. manokwari, and speculates that low winter temperatures may have restricted the invasion and establishment of P. manokwari in temperate countries.

Platydemus manokwari, like a number of other rhynchodemines of the Australia-New Guinea region, appears to be an upland species that naturally range from alpine through to sub-alpine, cool temperate and warm temperate zones to tropical climates. At the Pindaunde station on Mt. Wilhelm, New Guinea the mean daily temperature is 11.6°C, mean minimum of 4°C, absolute maximum of 16.7°C, absolute minimum of −0.8°C, and precipitation of some 3450 mm per year (Corlett, 1984), though it is expected that the microclimate on the sub-alpine forest floor would be milder. The climate at Pindaunde has been described as “wintery at night, (and) has days which seem to belong to a chilly spring or autumn” (McVean, 1968). Were P. manokwari introduced to temperate countries and escape hothouse or similar containment, the flatworm may well survive winters and become established. High frequencies of warm winters in temperate zones may also facilitate the establishment of the flatworm in these places (Sugiura, 2009). The flatworm has survived in the hothouse at Caen, and it is expected that it would also survive outdoors in this region, and even more easily in more southern part of Europe. An assessment of the global potential distribution of Platydemus manokwari, based on ecoclimatic data has not yet been undertaken.

Prey

Terrestrial molluscs form the principal prey upon which Platydemus manokwari has been observed to feed in the field (Table 2) and under laboratory and experimental conditions (Table 3), though the flatworm will also feed upon other soil-dwelling invertebrates including annelids, arthropods, nemerteans, and flatworms (Tables 2 and 3). The flatworm does not appear to be cannibalistic (Kaneda, Kitagawa & Ichinohe, 1990; Ohbayashi et al., 2005; Sugiura, 2010).

Table 2 Species reported as prey of Platydemus manokwari, in the field.

Platydemus manokwari has been recorded to feed mainly on land gastropod molluscs, and also on earthworms, insects and nemerteans.

Species	Location	Reference	
Mollusca: Gastropoda			
Achatina fulica Bowdich	Manokwari, Irian Jaya Agricultural Research Station	Hopper & Smith, 1992; Kawakatsu, Ogren & Muniappan, 1992; Kawakatsu et al., 1993; Mead, 1963; Mead, 1979; Muniappan et al., 1986; Raut & Barker, 2002; Schreurs, 1963; Sugiura, 2010; van Driest, 1968; Waterhouse & Norris, 1987; Winsor, 1999	
Euglandina rosea (de Férussac)	Okamura, Chichijima Island, Ogasawara Islands, Japan	Ohbayashi et al., 2005	
Partulidae: juvenile Partula, Streptaxidae: Gonaxis quadrilateralis (Preston),	Guam	Hopper & Smith, 1992	
Planorbidae: Physastra sp.	North Queensland, Australia urban	Waterhouse & Norris, 1987	
“Slugs” probably Vaginulidae	New Guinea	Winsor, Johns & Barker, 2004	
“Introduced predatory snails” probably Streptotaxidae: Gonaxis quadrilateralis, Incilaria sp. carcass	New Guinea, Chou-zan, Chichijima Island, Ogasawara Islands, Japan	Ohbayashi et al., 2005	
Annelida: Oligochaeta			
A pheretimoid earthworm	North Queensland, Australia urban	Waterhouse & Norris, 1987	
Haplotaxida spp. carcass	Komagari, Chichijima Island, Ogasawara Islands, Japan	Ohbayashi et al., 2005	
Arthropoda: Insecta			
Blattellidae: Calolampra sp.	North Queensland, Australia urban	Waterhouse & Norris, 1987	
Nemertea: Enopla			
Geonemertes pelaensis Semper	Chou-zan, Chichijima Island, Ogasawara Islands, Japan	Ohbayashi et al., 2005	
Vertebrata: Amphibia			
Litoria coerulea (White) carcass	North Queensland, Australia urban	Waterhouse & Norris, 1987	

Table 3 Species reported as prey of Platydemus manokwari, under laboratory conditions.

Platydemus manokwari is able to prey on a variety of gastropod molluscs, on nemerteans, earthworms and woodlice, and on other species of land planarians. All reports of prey refer to adults.

Species	Reference	
Mollusca: Gastropoda		
Partula sp.	Hopper, 1990	
Bradybaenidae: Acusta despecta sieboldiana (Pfeiffer), Bradybaena similaris (de Férussac), Euhadra amaliae callizona (Crosse), Euhadra peliomphala (Pfeiffer), Euhadra quaesita (Deshayes), Trishoplita conospira (Pfeiffer); Camaenidae: Satsuma japonica (Pfeiffer); Clausiliidae: Euphaedusa tau (Boettger), Pinguiphaedusa hakonensis (Pilsbry), Zaptychopsis buschi (Pfeiffer); Discidae: Discus pauper (Gould); Helicarionidae: Helicarion sp.; Limacidae: Lehmannia marginata (Müller); Ellobiidae: Pythia scarabaeus Linnaeus; Zonitidae: Zonitides arboreus (Say); Achatinidae: Achatina fulica	Kaneda, Kitagawa & Ichinohe, 1990; Kaneda et al., 1992	
Achatina fulica, Limax marginatus (Müller), Deroceras laeve (Müller), Euglandina rosea, Bradybaena similaris (de Férussac), Acusta despecta sieboldiana (Pfeiffer)	Ohbayashi et al., 2005	
Acusta despecta (Pfeiffer)	Sugiura, 2010	
Helicidae: Eobania vermiculata Müller	This paper	
Platyhelminthes: Tricladida		
Australopacifica sp., Bipalium kewense, Bipalium sp., Platydemus sp. 1; P. sp. 2	Ohbayashi et al., 2005	
Nemertea: Enopla		
Geonemertes pelaensis Semper	Ohbayashi et al., 2005	
Annelida: Oligochaeta		
Eisenia foetida Savigny	Sugiura, 2010	
Arthropoda: Crustacea		
Armadillidium vulgare Latreille	Sugiura, 2010	

A number of species of terrestrial flatworms will, when moisture conditions are right, seek prey above the ground. Platydemus manokwari has been observed feeding on both juvenile and adult partulid snails at heights above one metre in trees, and in captivity the flatworm fed on specimens of Partula sp. and Pythia sp. (Eldredge & Smith, 1995; Hopper & Smith, 1992). Experimentally, P. manokwari has been shown to track artificially created snail scent trails on the ground (Iwai, Sugiura & Chiba, 2010), and up trees, supporting the hypothesis that the introduction of P. manokwari is an important cause in the rapid decline or extinction of native arboreal snails as well as ground-dwelling snails on Pacific Islands (Sugiura & Yamaura, 2008).

Where there are sufficient individuals of P. manokwari following sensory cues of the same prey the flatworms can overwhelm their prey by sheer numbers in a gregarious or “gang” attack (Mead, 1963; Ohbayashi et al., 2005; Sugiura, 2010).

Waterhouse & Norris (1987) considered that P. manokwari appeared to be an opportunistic carnivore and generally unselective in the choice of prey. Success of Platydemus manokwari as a biological control agent for Achatina fulica can be attributed to its polyphagy, resistance to starvation, ability to survive and reproduce on alternative prey and potential to reproduce rapidly in synchrony with prey populations (Winsor, Johns & Barker, 2004).

Invasion of a site by Platydemus manokwari may directly and indirectly impact on native and introduced arboreal, terrestrial soil and to a much lesser extent semi-aquatic slow-moving invertebrate fauna.

Impacts

From an agricultural perspective Platydemus manokwari is not a direct plant pest. In fact it has been and probably will continue to be used by local farmers, and plant protection agencies in the Pacific region as a bio-agent in the control of outbreaks of the Giant African snail Achatina fulica (FAO-SAPA, 2002; Winsor, Johns & Barker, 2004), though other factors apart from flatworm predation may contribute to the decline in pest snail populations (Lydeard et al., 2004).

Examined from an environmental perspective, P. manokwari has demonstrably had a serious negative impact on the biodiversity of native snail populations in the Pacific region (Cowie, 2010) and wherever it is deliberately or accidentally introduced it will continue to pose a threat not only to native molluscs, but possibly to other slow-moving soil invertebrates (Sugiura, 2010). It may also indirectly have a negative impact on vertebrate species dependent upon these soil invertebrates.

An environmental pest risk assessment along the lines of that in the International Standards for Phytosanitary Measures (IPPC, 2004) may need to be undertaken for P. manokwari: an assessment of the probability of direct spread of the flatworm, considered by us to be low; whether the population is actively reproducing and is viable; an assessment of economic consequences, for example, potential threats to commercial snail farming; and environmental consequences, for example, negative impacts on soil invertebrate biodiversity in France and elsewhere. The extent of this incursion, and whether or not it is limited to the hothouse in Caen, the likely primary dispersal source of the current incursion, and possible secondary dispersal through plant exchanges between botanic gardens, and garden centres or plant supermarkets should also be considered.

Possible control options

As it is not a plant pest Platydemus manokwari is not listed in the European and Mediterranean Plant Organization A1 or A2 List of pests recommended for regulation as quarantine pests (EPPO, 2013a; EPPO, 2013b), nor listed by the European Alien Species Information Network (EASIN, 2014). In Europe, countries participating in the NOBANIS network have established a simple early warning system (Secretariat of NOBANIS, 2012). When a participating country becomes aware that a new alien species has been found in their country, a warning is sent to the other participating countries and posted on the NOBANIS website. This early warning enables countries to be alerted that a new species has been observed in the region.

Depending upon the outcome of an environmental risk assessment and related investigations, threats from Platydemus manokwari may need to be responded to in a similar manner to the invasive New Zealand flatworm Arthurdendyus triangulatus. This species is now subject to an EPPO Standard regarding import requirements (EPPO, 2000a) and nursery inspection, exclusion and treatment (EPPO, 2000b) for the flatworm (Murchie, 2010). The problem with P. manokwari is that even though it is primarily an environmental threat, it does not “indirectly affect plants through the effects on other organisms”. Consequently there is the possibility that responsibility for managing this invasive species may fall between the remits of agricultural and environmental regulatory bodies. This could delay effective management of P. manokwari.

Chemical control

Although a range of commercial pesticides were tested against Arthurdendyus triangulatus only gamma-HCH (Lindane), a broad spectrum, organochlorine insecticide gave significant control but was considered unsuitable for the widespread control of the flatworm (Cannon et al., 1999). There may be limited scope for the use of chemicals within an integrated approach to control of invasive alien flatworms combining chemical, physical and cultural methods (Blackshaw, 1996; Cannon et al., 1999).

Plant sanitization

Heat or hot water treatment of containerised plants that would kill invasive alien species has been investigated for Arthurdendyus triangulatus and Platydemus manokwari. Specimens of A. triangulatus were killed after immersion in a vial for 5 min in water at a temperature of 34°C (Murchie & Moore, 1998). This method showed great promise (Cannon et al., 1999) but it does not appear to have been used extensively; rather, the current advice to some amateur composters who had flatworm infestation was to place their compost in glasshouses to get the temperatures as high as possible before disseminating the compost (A Murchie, pers. comm., 2013). Similar experiments were undertaken on four invertebrate soil taxa that included Platydemus manokwari, using immersion in hot water at higher temperatures (Sugiura, 2008). It was found that exposure of the animals to hot water at ≥ 43°C to 50°C for 5 min resulted in 100% mortality for all species tested. In both sets of experiments the flatworms were tested in plastic vials. The ability of the hot water treatment to kill animals in potted soil masses was not examined. Depending upon its porosity and wetting ability the soil may act as a thermal buffer. A more promising method of hot-water treatment is the drenching method of Tsang, Hara & Sipes (2001) developed to sanitise potted plants of burrowing nematodes and potential other pest species. The treatment in which potted plants were drenched with hot water at 50°C for 15–20 min was more effective at killing burrowing nematodes than dipping potted plants in hot water for the same temperature-time regime. Based upon Sugiura’s (2008) data the temperature-time regime of the hot-water drench would kill Platydemus manokwari. The drench apparatus may be amenable to commercial development and use.

Biological control

As yet there are no known specific biological control methods for Platydemus manokwari. Terrestrial flatworms are considered to be top-level predators in the soil ecosystem (Sluys, 1999). Although nothing appears to be known about natural enemies of Platydemus manokwari, examples of predation on other species of land planarians by soil and associated fauna are known, mostly under laboratory conditions. They include an instance of predation of the Neotropical species Obama trigueira (E.M. Froehlich, 1955) by Enterosyringia pseudorhynchodemus (Riester, 1938) (Froehlich, 1956), and predation of five species of land planarians by P. manokwari (Ohbayashi et al., 2005). A chance field observation led to laboratory findings that Arthurdendyus triangulatus was eaten by larvae and adults of species of a carabid and a staphylinid beetles (Gibson, Cosens & Buchanan, 1997).

In a series of trials by Lemos, Canello & Leal-Zanchet (2012) the native Neotropical carnivorous mollusc Rectartemon depressus (Heynemann, 1868) was found to successfully predate upon specimens of at least 10 species of geoplanid terrestrial flatworms as well as five undescribed species of Geoplana, and also the introduced species Bipalium kewense. Whether other species of carnivorous molluscs successfully predate upon flatworms is not yet known. Platydemus manokwari predates upon at least two species of carnivorous molluscs observed in the field (Ohbayashi et al., 2005): the Rosy Wolf snail Euglandina rosea (de Férussac, 1821) and Gonaxis quadrilateralis (Preston, 1901); both these mollusc species were introduced in an attempt to control the Giant African snail Achatina fulica in the Pacific region (Davis & Butler, 1964; Lydeard et al., 2004).

Platydemus manokwari has a most unpleasant astringent taste (L Winsor, pers. obs., 1994), just as has been noted for other species (Dendy, 1891). Bellwood (D Bellwood, pers. comm. to LW, 1997) in his private urban garden, remarked that free-range domestic bantams that noticed P. manokwari on an upturned log pecked at, took the flatworms into their mouths, then immediately rejected them; when at a much later time P. manokwari was subsequently noticed by the bantams they refused to peck at the flatworms. This is similar to behaviour of domestic fowls offered Caenoplana spenceri (Dendy, 1891). Predation of flatworms by native species of birds has not been reported.

Predation of terrestrial flatworms by herpetofauna has also been investigated. The flatworm Bipalium adventitium Hyman, 1943, invasive in North America, was offered by Ducey et al. (1999) to six species of salamanders and two species of snakes; none of the herpetofaunal species tested treated Bipalium adventitium as a potential prey item.

Parasitization of P. manokwari by nematodes, gregarines or mycetophilid flies, known in other species of land planarians (Graff, 1899; Hickman, 1964), has not yet been observed.

Lemos, Canello & Leal-Zanchet (2012) advocate further experimental testing of other potential invertebrate and vertebrate predators of flatworms in an attempt to better understand predator–prey relationships, and cognisant of the risks associated with biological control, they consider the use of non-indigenous species should be avoided, and when necessary be based upon accurate pre-release testing and post-release monitoring.

Conclusion

The serious negative environmental impacts of Platydemus manokwari on the biodiversity of native land snails in the Indo-Pacific are well documented. The risks posed by the incursion of this species in France have not yet been assessed. The European Union has recently proposed new legislation to prevent and manage the rapidly growing threat to biodiversity from invasive species (European Commission, 2013). The proposal centres on a list of invasive alien species of concern for Europe, which will be drawn up with the Member States using risk assessments and scientific evidence. Whether or not Platydemus manokwari will be included on this list remains to be seen.

Supplemental Information

Supplemental Information 1 A complete French translation of the text: Le ver plat de Nouvelle-Guinée Platydemus manokwari en France, première mention en Europe: il faut agir maintenant

Click here for additional data file.

The authors are grateful to Gerard Peaucellier and Mrs J. Vivant for providing information on terrestrial flatworms in France, to David Bellwood in Townsville, Queensland for his observations on the feeding response of domestic bantams to P. manokwari, to Gilianne Brodie, School of Biological and Chemical Sciences, University of the South Pacific, Fiji, for her papers on the occurrence of Platydemus manokwari at Rotuma, to Archie Murchie, Agri-Food and Biosciences Institute, Belfast, Northern Ireland, for information and advice on current plant sanitation practices, standards and legislation for the Invasive Alien Species Arthurdendyus triangulatus, to Nicolas Puillandre (MNHN, Paris) for help with the sequences, to Pierre Lozouet (MNHN, Paris) who identified the prey snail. We are especially thankful to Damien Loisel and David Philippart (FREDON Basse-Normandie, France) and Martine Aires (DRAAF Basse-Normandie, France) for collection of specimens and administrative support, and to Damien L’Hours and Nelly Hubert (Ville de Caen, France) for allowing collection in the hothouse in Caen and providing administrative support. Hugh Jones, Ana M. Leal-Zanchet, Fernando Carbayo and Maria Riutort are acknowledged for their constructive comments on an earlier version of this paper.

Additional Information and Declarations

Competing Interests

Author Contributions

Data Deposition

Jean-Lou Justine is an Academic Editor for PeerJ. Jean-Lou Justine, Delphine Gey and Jessica Thévenot are employees of the Muséum National d’Histoire Naturelle.

Jean-Lou Justine conceived and designed the experiments, performed the experiments, analyzed the data, contributed reagents/materials/analysis tools, wrote the paper, prepared figures and/or tables, reviewed drafts of the paper, initiated citizen science program about land planarians in France.

Leigh Winsor conceived and designed the experiments, performed the experiments, analyzed the data, wrote the paper, prepared figures and/or tables, reviewed drafts of the paper, provided unpublished information.

Delphine Gey performed the experiments, analyzed the data, contributed reagents/materials/analysis tools, reviewed drafts of the paper, performed molecular analyses.

Pierre Gros performed the experiments, contributed reagents/materials/analysis tools, prepared figures and/or tables, reviewed drafts of the paper, made the photographs.

Jessica Thévenot analyzed the data, reviewed drafts of the paper, provided administrative contacts and information concerning regulation of invasive species, made maps.

The following information was supplied regarding the deposition of related data:

GenBank KF887958.

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
