# Peer review of "The invasive New Guinea flatworm Platydemus manokwari in France, the first record for Europe: time for action is now"

_PeerJ, doi:10.7717/peerj.297_

## Round 0.1 · original submission · Minor Revisions

Please consider carefully the modifications requested by the different referees that should allow you to resubmit a final version that could be acceptable

·

Basic reporting

The manuscript addresses a relevant issue and is properly written. I think some parts of the paper could be improved (please see comments below). Thus, I suggest the acceptance of the paper after minor revision.

Abstract
I suggest introducing the issue by mentioning the existence of various exotic species of land flatworms in Europe and its possible threat.
I would more directly say that a non-native species of land flatworm, collected in Caen (France), was identified as the New Guinea flatworm Platydemus manokwari.
I suggest stating the aims of the article subsequently.

Introduction
The table 5.1 mentioned on line 25 is a table of Winsor´s paper, but I think it is unnecessary to mention it.

Material and methods
The authors state that prey experiments were undertaken but their results were not mentioned. Thus, I suggest adding comments in the results about this topic or deleting the sentences in lines 73-74 and removing figure 5. Table 3 indicates the predation of Eobania vermiculata, but I could not find these results in the text.

Results
l. 98: the figure 1 also should be mentioned here.
l. 104-105: is there anything to say about the morphology of the pharynx?
See comments about the predation behaviour above.

Discussion
l. 215-216: In my opinion, the sentence “It is probable that most vertebrates would find the flatworm distasteful.” is very speculative. Taste may vary too much. I suggest removing it.

l. 315-317: This topic was poorly developed. How about the natural enemies of other land planarians?

l. 266: I would delete “Statutes, Standards and Guidelines.”

I was not able to find mention of table 3 in the text.

Figures
Figure 1: please indicate the median stripe. I suggest the following title: Specimen of Platydemus manokwari de Beauchamp, 1963 collected in a hothouse, Caen, France, in dorsal view.
Figure 2: please indicate the eyes in this figure. I also suggest the following title: Detail of the head of a specimen of Platydemus manokwari de Beauchamp, 1963 collected in a hothouse, Caen, France, in lateral view.
Figures 3 and 4: the text of the legends could be tighter and the main features should be indicated in the figures.
Figure 4 (legend): correct “millimeters”

Table titles:
Table 1: I think the title should be Distribution records of Platydemus manokwari. The subtitle (“Platydemus manokwari has been recorded …”) is not necessary.

Tables 2 and 3: The subtitles are not necessary.

Experimental design

No aditional comments.

Validity of the findings

No aditional comments.

Additional comments

See under "Basic Reporting"

·

Basic reporting

An important and timely paper and acceptable subject to very minor alterations as suggested below.

Experimental design

Not relevant.

Validity of the findings

Fully justified.

Additional comments

Suggested amendments:
Each species requires a naming authority and date at its first mention in the text. E.g. lines 34, 42, 43, 49, 50, others in the discussion and tables.

Line 50: This implies that C. coerulea has been found in Europe only in France and Menorca. But it has previously been found at one locality in the UK.

Platydemus manokwari should be written out in full at first mention and if starting a sentence (a sentence should not start with an abbreviation) but as P. manokwari if within a sentence.

Line 94. In expressing measurements, a space is required between a number and its relevant units, i.e. 50 mm not 50mm.

Line 234. Delete “Waterhouse & Norris” within the brackets.

·

Basic reporting

A new record of a potentially invasive species to Europe is presented in the manuscript. Some specimens were identified as P. manokwari and abundant data from literature on biology of the species and pest control are provided. The manuscript is a relevant scientific contribution and should be published after minor revisions.

In the following I summarize the main comments. These comments and additional suggestions, corrections and comments are found along the text as well (see attached pdf document).

Experimental design

Please see General Comments for the Author

Validity of the findings

Please see General Comments for the Author

Additional comments

The Abstract and the Introduction are concise and informative. The figures are of high quality and informative. The three tables are also informative. I am not English native speaker, but some phrases, chiefly in the Discussion could seemingly be improved.

The title does not focus on the long section dedicated to biological aspects and pest control; the title could be accordingly adapted (something like "First record of invasive New Guinea flatworm Platydemus manokwari in Europe: time for action is now")

The authors consider all exotic land planarians as "invasive alien species". This might not be true; there is some debate on terminology to ecologically define exotic species (for instance, Coulatti & McIsaac 2004. Diversity and Distributions 10: 135–141). After the definition of IAS provided by the Convention on Biological Diversity (http://www.cbd.int/idb/2009/about/what/) some land planarians would not macht be considered IAS. Indeed, B. kewense, D. striata seemingly are synanthropic species - at least in the Neotropic- and do not invade natural habitats. Thus the authors should adopt a definition along the text.

The specimens analysed are not unequivocally assigned a code, not in the pictures, nor their DNA sequences, not the morphological description. This point is very important since authors trust the identification of the specimens on a combination of molecular and morphological data, but do exclude morphology of internal organs, namely that of the copulatory apparatus. By omitting proper coding for the specimens, authors will hamper an eventual taxonomic revision of the specimens. Moreover, cryptic land planarian species are also known. Thus, it wolud be very welcome to explicitely indicate which specimen was described (DNA, morphology). It is also not clear how many voucher specimens are deposited in the MNHN.

In the Discusion the authors have followed the Standards Sections of the Manuscript Organization suggested by PeerJ. However, the Discussion is actually composed of two sections: a proper discussion on the identification of the specimens and a review on biology and control of the invasive species. However, I have no suggestion for a different organization of the sections.

---

## Round 0.2 · accepted · Accept

Thanks for considering efficiently the modifications requested by the referees.